# Enteric-Coated Cologrit Tablet Exhibit Robust Anti-Inflammatory Response in Ulcerative Colitis-like In-Vitro Models by Attuning NFκB-Centric Signaling Axis

**DOI:** 10.3390/ph16010063

**Published:** 2022-12-31

**Authors:** Acharya Balkrishna, Rani Singh, Vivek Gohel, Sagar Arora, Rishabh Dev, Kunal Bhattacharya, Anurag Varshney

**Affiliations:** 1Drug Discovery and Development Division, Patanjali Research Institute, Uttarakhand, Haridwar 249 405, India; 2Department of Allied and Applied Sciences, University of Patanjali, Patanjali Yog Peeth, Uttarakhand, Haridwar 249 405, India; 3Patanjali Yog Peeth (UK) Trust, 40 Lambhill Street, Kinning Park, Glasgow G41 1AU, UK; 4Vedic Acharya Samaj Foundation Inc., NFP 21725 CR 33, Groveland, FL 34736, USA; 5Special Centre for Systems Medicine, Jawaharlal Nehru University, New Delhi 110 067, India

**Keywords:** ulcerative colitis, enteric-coated tablet, herbal medicine, anti-inflammatory, anti-oxidant, inflammatory bowel disease

## Abstract

Ulcerative colitis (UC) is an inflammatory bowel disease that affects the patients’ colorectal area culminating in an inflamed ‘leaky gut.’ The majority of UC treatments only provide temporary respite leading to its relapse. Therefore, this study investigated the efficacy of the enteric-coated ‘Cologrit’ (EC) tablet in alleviating UC-like inflammation. Cologrit is formulated using polyherbal extracts that have anti-inflammatory qualities according to ancient Ayurveda scriptures. Phytochemical profiling revealed the presence of gallic acid, rutin, ellagic acid, and imperatorin in Cologrit formulation. Cologrit treatment decreased inflammation in LPS-induced transformed THP-1 macrophages, and TNF-α-stimulated human colorectal (HT-29) cells through the modulation of NFκB activity, IL-6 production, and NFκB, IL-1β, IL-8, and CXCL5 mRNA expression levels. Cologrit also lessened human monocytic (U937) cell adhesion to HT29 cells. Methacrylic acid-ethylacrylate copolymer-coating of the enteric Cologrit tablets (EC) supported their dissolution, and the release of phytochemicals in the small intestine pH 7.0 environment in a simulated gastrointestinal digestion model. Small intestine EC digestae effectively abridged dextran sodium sulfate (2.5% *w*/*v*)-induced cell viability loss and oxidative stress in human colon epithelial Caco-2 cells. In conclusion, the enteric-coated Cologrit tablets demonstrated good small intestine-specific phytochemical delivery capability, and decreased UC-like inflammation, and oxidative stress through the regulation of TNF-α/NFκB/IL6 signaling axis.

## 1. Introduction

Ulcerative colitis (UC) is an idiopathic chronic inflammatory bowel disease of the colon and rectum. The disease’s aetiology is mostly unknown, but it is frequently linked to genetic, dietary, lifestyle, microbial infection, and environmental factors, alone or in combination [1,2]. Adults between the ages of 30–40 are mostly affected by UC, which is characterized by relapsing and remitting mucosal inflammation [3]. Patients with inflammatory bowel syndrome-like UC show elevated levels of pro-inflammatory cytokines, chemokines, and adhesion molecules in their intestinal mucosa biopsies [4,5,6,7,8]. Individual patients’ UC affects different regions of the colon, typically beginning in the rectum and progressing to the proximal segments. The disease can spread throughout the colon (known as pan-colitis), or be localized between the proximal and rectal regions of the colon (ulcerative proctitis) [9,10]. Topical, oral, and systemic administration of aminosalicylates, immunosuppressants, and steroids are currently used to treat UC [3]. In the majority of acute UC patients, intravenous corticosteroids and rescue therapy with infliximab are effective in suppressing the disease. Nearly one-fifth of responders still undergo colectomy due to relapse [11].

The intestinal innate immune system recognizes and responds to pathogen-associated danger signals (PAMPs)/damage-associated danger signals (DAMPs) during the onset of UC by initiating a pro-inflammatory cascade mostly centering through the NFκB pathways [12,13,14]. This stimulation damages the mucosal and epithelial tissue of the intestine and disrupts homeostasis. The chronicity of UC-related inflammation causes patients to become resistant to anti-inflammatory drugs on multiple occasions. Furthermore, the long-term use of immunosuppressive drugs may result in unintended health consequences. In this context, Ayurveda poly-herbal medicines can provide multifaceted relief to UC patients through prophylaxis and therapeutic medications [15]. ‘Cologrit’ is an Ayurveda polyherbal medicine made from *Aegle marmelos* (L.) Corrêa, *Holarrhena antidysenterica* (L.) Wall. ex A. DC, *Cuminum cyminum* L., *Trachyspermum ammi* (L.) Sprague, *Foeniculum vulgare* Mill., *Rosa indica* L., and *Cinnamomum camphora* (L.) J. Presl (see Table 1). These herbs have been linked to anti-inflammatory properties in various disease conditions in ancient Ayurveda texts [16]. Active phytochemical ingredients present in *Aegle marmelos* (L.) Corrêa fruit extract were found to modulate inflammation-associated responses in inflammatory bowel syndrome (IBD) [17]. Similarly, *Holarrhena antidysenterica* (L.) Wall. ex A. DC treatment prompted full recovery in chronic UC patients in a randomised clinical trial with no further relapses [18]. *Cuminum cyminum* L., *Trachyspermum ammi* (L.) Sprague, and *Foeniculum vulgare* Mill. have also been shown to have anti-inflammatory effects in the gastrointestinal tract ailments [19,20]. Although individual herbal extracts have been found to protect against ulcerative colitis, as well as related inflammation and oxidative stress, their synergistic action in combination has yet to be investigated.

The easiest oral route delivery of medicines is in the form of tablets. The term “tablet” refers to a pharmaceutical dose made up of active ingredients and excipients. Polymer coatings present on tablets improves their appearance, easier swallowing, and controls site-specific release of active ingredients. The enteric coating (EC) of the surface inhibits solubility of the tablet in the low pH conditions of the stomach, while supporting dissolution under neutral pH condition of the intestinal region [21]. Methacrylic acid—ethylacrylate copolymer (1:1), Type A (MMA) is an anionic polymer used as an EC on tablets. It is also referred to as “Methacrylic Acid Copolymer, Type C” in USP XXI/NF XVI [22]. MMA coating supports solubilization of the tablets at pH greater than 5.5 and preventing enzymatic digestion of the active ingredients [22].

In this study, the effectiveness of EC Cologrit was analyzed for its site-specific delivery of active phytochemical and UC-like inflammation prevention using human colonic HT-29 and Caco-2 cells. The Cologrit phytochemical profile was determined using the High-Performance Liquid Chromatography (HPLC) method. Anti-inflammatory activity of the Cologrit was studied in inflamed HT-29 and THP-1 cells for the amelioration of pro-inflammatory cytokines and chemokines. NFκB activation was analyzed through mRNA expression studies in HT-29 cells as well as through THP-1-based reporter cell assay. The EC Cologrit tablets were evaluated for their hardness, friability, dimensions, and pH-based dissolution using an in vitro enzymatic digestion model. The release of phytochemicals from intestinal dissolution of the tablets was tested using HPLC analysis and their efficacy was tested in dextran disodium sulfate (DSS)-simulated inflammation model of human colon Caco-2 cell.

## 2. Results

### 2.1. Cologrit Phytochemical Profiling and Tablet Preparation

HPLC analysis of the Cologrit showed the presence of Gallic acid (retention time 10.29 min): 0.19 µg/mg; Rutin (retention time 29.61 min): 0.34 µg/mg; Ellagic acid (retention time 30.60 min): 0.39 µg/mg; and Imperatorin (retention time 68.09 min): 1.53 µg/mg (Figure 1; Table 2). Based on literature review, the origin of Gallic acid was traced in all the phytocomponents of Cologrit formulation, that is *Aegle marmelos* (L.) Corrêa, *Holarrhena antidysenterica* (L.) Wall. ex A. DC, *Cuminum cyminum* L., *Trachyspermum ammi* (L.) Sprague, *Foeniculum vulgare* Mill., *Rosa indica* L., *Cinnamomum camphora* (L.) J. Presl. [19,23,24,25,26,27,28,29,30,31,32,33] (Table 2). Presence of Rutin in Cologrit has been traced in *Aegle marmelos* (L.) Corrêa, *Holarrhena antidysenterica* (L.) Wall. ex A. DC, *Cuminum cyminum* L., *Trachyspermum ammi* (L.) Sprague, and *Cinnamomum camphora* (L.) J. Presl. [26,27,31,33,34,35] (Table 2). Ellagic acid has been detected in *Aegle marmelos* (L.) Corrêa, *Cuminum cyminum* L., *Foeniculum vulgare* Mill., and Imperatorin has been identified in plants *Aegle marmelos* (L.) Corrêa, and *Foeniculum vulgare* Mill. [19,25,29,31,36,37]. Cologrit tablets were made using 381 mg of Cologrit powder mixed with 159 mg of excipients (Table 3). Tablets were EC with a film containing 6% (*w*/*w*) MMA. The dry Cologrit tablets had a diameter of 6.43 ± 0.01 mm, a thickness of 10.86 ± 0.06 mm, an average tablet had a weight of 542 ± 3.20 mg, hardness of 6.96 ± 0.15 Kg/cm^2^, and a friability of <1% (Table 4).

### 2.2. Anti-Inflammatory and Anti-Adhesion Activity of Cologrit

Bacterial lipopolysaccharide (LPS) binds to the toll-like receptor four in macrophages activating NFκB-associated cell signaling pathways, and prompt inflammation. Inflamed macrophages release pro-inflammatory cytokines causing breakdown of the epithelial barrier, resulting in ‘leaky gut’ [39]. The stimulation of the PMA-transformed THP-1 macrophages with LPS (100 ng/mL) resulted in considerable (*p*-value < 0.001) TNF- α (410 ± 15.3 pg/mL) and IL-6 release (450 ± 5.5 pg/mL) compared to untreated control (TNF- α- not detected; IL6- 4.8 ± 1.4 pg/mL) (Figure 2A,B). Transformed THP-1 showed a significant decrease in the LPS-stimulated release of TNF-α, when treated with Cologrit at the concentration of 100 µg/mL: 225 ± 16 pg/mL (Figure 2A). However, no response was observed at the lower treatment doses of Cologrit. Pre- and co-treatments of the transformed THP-1 with Cologrit also significantly (*p*-value < 0.01) reduced the release of soluble IL-6 cytokine (30 µg/mL: 312.1 ± 50.5 pg/mL and 100 µg/mL: 240.4 ± 25.7 pg/mL) (Figure 2B). The release of soluble TNF-α is performed by inflamed monocytes, macrophages, platelets, adipocytes, and T cells during prevailing UC conditions [40]. This pro-inflammatory cytokine binding to its cell surface receptor (TNFR) activating NFκB signaling in the gut epithelial and endothelial cells. This results in the expression of pro-inflammatory cytokines, chemokines and adhesion molecules in these cells, and the recruitment of immune cells to the site of injury causing edema and granuloma formation [14,40,41]. THP-1-Blue NFκB reporter cells stimulated with TNF-α (10 ng/mL) showed a significant (*p*-value < 0.001) induction of NFκB (13.15 ± 2.4-fold) detected through SEAP activity compared to the untreated control cells (Figure 2C). Subsequent co-treatment with the Cologrit showed a concentration-dependent reduction in the TNF-α stimulated NFκB-related SEAP activity (10 µg/mL: 9.94 ± 1.8 folds; 30 µg/mL: 8.83 ± 1.94 folds (*p*-value < 0.05); 100 µg/mL: 7.21 ± 1.41 (*p*-value < 0.01)) compared to untreated control (Figure 2C).

Cologrit was found to be biocompatible in the HT-29 cells and capable of substantially (*p*-value < 0.01) enhancing cell viability in HT29 cells (Figure 3A). LPS induced TNF-α controls the expression of pro-inflammatory cytokines, chemokines, and adhesion molecules by regulation of the NFκB pathway [8,42,43,44,45]. The TNF-α (10 ng/mL) stimulation of HT-29 cells resulted in a substantial increase (*p*-value < 0.05) in mRNA expression levels of the NFκB (3.33 ± 1.66 folds), IL-1β (5.82 ± 2.06 folds), IL-8 (4.37 ± 2.01 folds), and CXCL5 (1.88 ± 0.70 folds) (Figure 3B–E).

The treatment of the TNF-α-inflamed HT29 cells with Cologrit significantly reduced the upregulated mRNA expressions for NFκB, IL-1β, IL-8, and CXCL5 (Figure 3B–E). Maximum reduction (*p*-value < 0.01) for all the TNF-α stimulated mRNA expressions was observed at the Cologrit concentration of 100 µg/mL (NFκB (1.33 ± 0.54 folds), IL-1β (1.52 ± 1.08 folds), IL-8 (1.19 ± 0.66 folds), and CXCL5 (0.62 ± 0.15 folds) (Figure 3B–E). The attachment of the immune cells to the epithelial lining of the gut aggravate UC by progressively deteriorating the intestinal barrier [46]. The TNF-α (10 ng/mL) treatment of the HT-29 cells significantly (*p*-value < 0.001) increased the adherence of U937 cells to the surface of HT-29 cells as observed through a 1.35 ± 0.15-fold increase in fluorescence (Figure 3F). The pre- and co-treatment of HT-29 cells with the Cologrit resulted in a significant (*p*-value < 0.01) dose-dependent reduction (10 µg/mL: 1.24 ± 0.16-fold; 30 µg/mL: 1.10 ± 0.08-fold; 100 µg/mL: 1.06 ± 0.13-fold) in U937 cell adherence-associated fluorescence in the HT-29 cells (Figure 3F).

### 2.3. In Vitro Release of Phytocontituents from EC Cologrit Tablet

EC Cologrit tablets were subjected to in vitro gastrointestinal digestion, which included passing through the mouth (15 s), stomach (2 h), and intestine (2 h) [47]. The EC tablets did not show any evidence for the EC solubilization, or cracking in mouth or stomach phases (Figure 4A,B). In the intestinal phase, the MMA-coating ruptured within 20 min of treatment exposing the tablet interior causing a complete dissolution within 120 min (Figure 4C).

UV spectroscopy analysis of the solubilized EC Cologrit tablets showed the presence of a specific absorption band between 307 and 371 nm, with a ‘λ maxima’ (λmax) at 339 nm with no interferences (Figure 5A,B). No Cologrit spectra was detected in the stomach phase ‘chyme’ between t = 0 and t = 120 min (Figure 5A). However, a time-dependent absorbance spectrum for Cologrit tablet was observed in the intestinal phase ‘digesta’ indicating its dissolution (Figure 5B). The EC Cologrit tablet disintegrated fastest between 40–60 min, after which the process slowed, indicating that the EC layer dissolved swiftly in the simulated digestive juice (pH = 6.8) (Figure 5B). The findings were validated by HPLC analysis, which revealed the lack of Gallic acid, Rutin, Ellagic acid, and Imperatorin in stomach phase ‘chyme’ samples (t = 0, t = 60, and t = 120) (Table 4, Figure 5C). HPLC analysis of the intestinal phase revealed the presence of Gallic acid: t = 60 min (0.09 ± 0.06 µg/mg) and t = 120 min (0.12 ± 0.05 µg/mg); Rutin: t = 60 min (0.25 ± 0.02 µg/mg) and t = 120 min (0.29 ± 0.0 µg/mg); Ellagic acid: t = 60 min (0.21 ± 0.04 µg/mg) and t = 120 min (0.28 ± 0.07 µg/mg); and Imperatorin: t = 60 min (0.04 ± 0.02 µg/mg) and t = 120 min (0.07 ± 0.02 µg/mg) (Table 4). When the HPLC profiles of EC Cologrit tablets disintegrated in the intestinal phase (t = 120 min) was compared to that of undigested Cologrit alone profile, the phytochemical release efficacy was determined to be 63.15% for Gallic acid, 85.29% for Rutin, 71.79% for Ellagic acid, and 4.57% for Imperatorin (Figure 5; Table 4). The low detectable quantity of Imperatorin may be due to its conversion into xanthotoxol, which was not investigated in this work [48].

The efficacy of the GIT digested EC Cologrit tablet was analyzed in the human colonic Caco-2 cells. Digestae of the EC Cologrit tablet was found to be biocompatible in Caco-2 cells up to the tested concentration of 100 µg/mL and 24 h treatment time (Figure 6A). The treatment of the Caco-2 cells with DSS caused a concentration-dependent loss of cell viability (Figure 6B). DSS showing an inhibitory concentration 50% (IC_50_) of 3.3% (*w*/*v*) (Figure 6B). Hence for the study the treatment doe of 2.5% (*w*/*v*) was selected. Pre- and co-treatment of the Caco-2 cells with the undigested Cologrit showed a minor protection against the 2.5% (*w*/*v*) DSS-induced loss of cell viability (0 µg/mL: 64.27 ± 0.58%; *p*-value < 0.001).

Statistically significant (*p*-value < 0.05) inhibition of DSS-induced cell viability loss by undigested Cologrit was observed at the 100 µg/mL concentration (79.38 ± 3.98%) (Figure 6C). In the DSS-treated Caco-2 cells, pre- and co-treatment with Cologrit tablet digestae significantly (<0.001) inhibited loss of cell viability in a dose-dependent manner (1 µg/mL: 74.62 ± 3.70%; 3 µg/mL: 78.60 ± 1.26%; 10 µg/mL: 82.03 ± 2.43%; 30 µg/mL: 84.32 ± 0.67%; 100 µg/mL: 91.54 ± 0.83%) (Figure 6D). 2.5% (*w*/*v*) DSS treatment also triggered oxidative stress in the Caco-2 cells as observed through an increase in ROS levels (138.99 ± 23.87%; *p*-value < 0.06). Pre- and co-treatment of the Caco-2 cells with the digestae of Cologrit tablet showed a significant (*p*-value < 0.001) reduction in the DSS-induced ROS levels (1 µg/mL: 66.44 + 20.75%; 3 µg/mL: 71.78 ±17.64%; 10 µg/mL: 99.48 ± 10%; 3 µg/mL: 85.45 ± 2.81%; 100 µg/mL: 73.21 ± 9.09% (Figure 6E). Hence, the intestinal phase Cologrit tablet digestae showed good anti-oxidant property.

The overall findings validated EC Cologrit tablets to have good anti-inflammatory and anti-oxidant qualities. It reduced production of pro-inflammation cytokines in LPS stimulated macrophages and suppressed NFκB activity induced by TNF-α treatment. EC supported a site-specific delivery of the Cologrit phytochemicals in the intestinal region without getting digested in the high pH of stomach. Biological efficacy of the Cologrit phytochemicals present in the intestinal digestae were not affected by the presence of digestive enzymes and ions. The phytochemical containing Cologrit digesta readily reduced UC like cellular damages in the DSS-treated Caco-2 cells.

## 3. Discussion

Ulcerative colitis is an inflammatory illness that causes inflammation and pain in the gastrointestinal tract. The condition produces severe discomfort and requires drastic lifestyle modifications. UC symptoms include ‘*Shula*’ (abdominal discomfort), ‘*Gudapaaka*’ (rectum burning), and ‘*Trishna*’ (extreme thirst), according to the ancient Ayurveda text-book ‘Charak Samhita’ [16]. Corticosteroids (CS) form a major part of UC medication, which attenuates the immune response giving relief in the form of anti-inflammatory drugs [49]. CS is administered to UC patients either orally or systemically and their long-term use can cause osteoporosis, depression, moon face, type 2 diabetes, and cataracts [1,49]. Because of its simplicity, and convenience, drug administration through the oral route is considered the best therapeutic option [50]. Oral drugs, on the other hand, are subjected to a harsh environment specifically during ongoing inflammatory bowel disease conditions. The orally delivered medicines may get degraded through hydrolysis, or oxidation in the presence of a variety of digestive enzymes in the stomach and small intestine regions. The medicines may also get denatured at varying pH conditions between acidic (pH 1–3) present in the stomach and neutral to slightly alkaline (pH 6–7.5) present in the duodenum, jejunum, and ileum [51]. As a result, pH-triggered release mechanisms are widely used in oral administration to improve the stability and controlled release of medications in the gastrointestinal region. In the present study, we focus on pH-responsive polymeric EC Cologrit tablets for oral delivery of anti-inflammatory and anti-oxidant phytochemicals to the UC specific regions of colon.

The polyherbal Cologrit is based on the Ayurveda recommended application of Bel (*Aegle marmelos* (L.) Corrêa), Kutaj (*Holarrhena antidysenterica* (L.) Wall. ex A. DC), Jeera (*Cuminum cyminum* L.), Ajwain (*Trachyspermum ammi* (L.) Sprague), Sounf (*Foeniculum vulgare* Mill.), Gulab (*Rosa Indica* L.), and Kapur (*Cinnamomum camphora* (L.) J. Presl). Many of these individual plant extracts have been applied for the remediation of IBD and inflammation in general [17,18,52,53,54]. Gallic acid, Rutin, Ellagic acid, and Imperatorin were identified in the Cologrit. Gallic acid and Ellagic acid have been shown to reduce UC-like inflammation by inhibiting pro-inflammatory cytokines via the P65-NFκB, IL-6/*p*-STAT3, iNOS, and COX-2 pathways, as well as increasing anti-inflammatory cytokines such as IL-4 and IL-10 in the intestinal mucosa [55,56,57]. Imperatorin acts as an anti-inflammatory mediator by reducing the expression of Nrf-2, ARE, and HO-1, as well as suppressing TNF-α, IL-6, and related NFκB activation [58,59].

In our study, Cologrit was found to be biocompatible towards the intestinal epithelial cells. The medicine showed anti-inflammatory properties against bacterial lipopolysaccharide driven production of soluble TNF-α and IL-6 in the PMA-transformed THP1 macrophages. NFκB is a nuclear factor that plays a pivotal role between the stimulants and the pro-inflammatory cytokine release [60]. TNF-α signaling controls the expression of pro-inflammatory cytokines, chemokines, and adhesion molecules such as intercellular adhesion molecule-1 (ICAM-1) and vascular cell adhesion molecule-1 (VCAM-1) through the NFκB pathway [42,43,44,45]. TNF-α interacts to the cell surface receptor TNFR1 and stimulates the recruitment of the adaptor proteins TRADD and Receptor-interacting serine/threonine-protein kinase 1 (RIP1) [61]. TRADD recruits TNFR-associated factor (TRAF) 2, as well as ubiquitin ligases, which promote RIP1 ubiquitination and the formation of a scaffold composed of ubiquitinated RIP1, NFκB essential modulator (NEMO), TRADD, and TNFR1 with linear polyubiquitin chains. This newly formed scaffold promotes downstream phosphorylation and degradation of the NFκB inhibitory IκB-α protein activating NFκB existing in the cytoplasmic pool [41,61]. Activated NFκB translocate to the cell nucleus and initiate the synthesis of pro-inflammatory cytokines.

In the present study, the anti-inflammatory activity of the Cologrit was found to be NFκB centric. TNF-α treatment controlled the release of NFκB from the THP-1–Blue NFκB reporter cells that was abolished by treatment with Cologrit. Similarly, TNF-α stimulated the mRNA expression of NFκB in the HT-29 cells that was reduced through treatment with Cologrit. Cologrit treatment also reduced the TNF-α prompted mRNA expression of pro-inflammatory cytokines and chemokines IL-1β, IL-8, and CXCL5. As a consequence, the reduction of IL-1, IL-8 in the HT-29 cells can be connected to downregulation of NFκB mRNA expression in the current study. *Aegle marmelos* (L.) Corrêa one of the main components of Cologrit has been found to reduction the mRNA expression of IL-6, TNF-α, TNF receptor-1 (TNFR1), TNFR1-associated death domain protein (TRADD) and NFκB [62,63,64]. Gallic acid, Imperatorin, Umbelliferon and Lupeol (triterpenoid) present in *Aegle marmelos* (L.) Corrêa and Cologrit are known to lowering TNF-α stimulated NFκB expression in various cell types by interfering with cell signaling pathway [64,65,66]. As a result, in the current investigation, suppression of IL-1, IL-8, as well as soluble TNF-α and IL-6 mRNA expression, can be linked to downregulation of NFκB mRNA expression. NFκB holds importance as a therapeutic target for controlling inflammation in UC [14]. Hence our results hold significance in the stherapeutic treatment of UC by controlling inflammation through TNF-α/NFκB signaling pathway modulation.

CXCL5, also known as epithelial cell-derived neutrophil activator 78 (ENA-78), is a pro-inflammatory chemokine that is produced by epithelial cells and has a role in neutrophil activation [67,68]. CXCL5, or ENA78, is significantly detected in colonic mucosa biopsy samples from UC patients [69,70]. TNF-α stimulates CXCL5 transcription via NFκB activation [71]. CXCL5 mRNA expression was found to be increased in the TNF-α stimulated HT-29 cells. This was reduced by treating the inflamed cells with Cologrit. TNF-α stimulates activation of NFκB through induction of IκB-α proteasomal degradation has been found to play a major role in the activation of adhesion molecules such as ICAM-1 [72]. The deactivation of the NFκB by blocking of TNF-α induced IκB-α proteasomal degradation has been found to reduce the production of adhesion molecules in the intestinal epithelial cells thereby reducing adhesion of immune cells [72]. In our study, Cologrit treatment inhibited the binding of TNF-α stimulated binding of U937 cells to the surface of HT-29 cells. This indicated a reduction in the production of adhesion molecules in the TNF-α induced HT-29 cells following treatment with Cologrit.

EC are designed to save the active ingredients of the tablets from dissolution and digestion within stomach under high acidic pH conditions [73]. Previous studies have shown that the Methacrylic acid—ethyl acrylate copolymer protects the dissolution of active ingredients under acidic conditions [74,75]. In the present study, we synthesized Cologrit EC tablets using MMA polymer showing integrity of coating at 6% (*w*/*w*) concentration. Using an in vitro GIT digestion model along with UC and HPLC analysis, we showcased the release of tablet ingredients at pH 7.0 and strong release phytochemicals. The controlled release of the phytochemicals Gallic acid, Ellagic acid and Rutin at the site of UC inflammation would significantly enhance their biological activity, as clearly seen in DSS stimulated Caco-2 cells showing cell protection against loss of cell viability and oxidative stress. Additionally, Gallic acid is known to holistically protect the probiotic gut microbiome population and eliminate pathogenic species like Firmicutes and Proteobacteria phyla through the amendment of carbohydrate, bile acid and amino acid metabolisms [76]. Additionally, Ellagic acid and Rutin are known to support the anti-inflammatory activity of the gut against the inflammation produced in UC regions [57,77]. Hence, their delivery at the site of disease without degradation in the stomach region by the EC tablet would enhance their biological efficacy in remediation of UC like inflammatory diseases.

Taken together, the study showed an efficacy of the polyherbal Cologrit and tablet in reducing inflammation, the loss of cell viability, and oxidative stress in the colonic epithelial cells. The anti-inflammatory activity of the Cologrit was identified as NFκB activation centric through which it controls the expression and release of pro-inflammatory cytokines. EC of the Cologrit tablets enhanced their site specific efficacy by protection against dissolution in high acidic pH of the stomach region. The EC tablets showed the same efficacy as the Cologrit, indicating no interference in the phytochemical release and biological efficacy from the EC polymer MMA. All of the plant chemicals present in the polyherbal Cologrit originating from the different plant components work tandemly in controlling gastrointestinal inflammation and oxidative stress. However, their individual role in the formulation, still need to be explored.

## 4. Conclusions

Ultimately, the study showcased the anti-inflammatory behavior of the enteric-coated Cologrit tablet in UC-like disease model. The mode of action for the Cologrit in the inflamed cells was found to be through the controlled NFκB activation by inducers such as TNF-α. Cologrit was found to function in the same way in both the macrophages and gut epithelial cells. The formation of EC Cologrit tablets enhanced their site specific activity through the inhibition of its dissolution in acidic pH conditions existing in stomach region. However, the tablets were found to be readily dissolved in the intestinal neutral pH, delivering its phytochemicals and showing efficacy in ameliorating UC-like cell damage and oxidative stress. Overall, Cologrit has been shown to be a unique herbal medicine for the control of ulcerative colitis, such as inflammatory bowel syndrome. In future studies, the plant components of the polyherbal Cologrit formulation could well be examined for their individual role in managing the various aspects of ulcerative colitis associated inflammation.

## 5. Materials and Methods

### 5.1. Reagents

Raw materials for the herbal components were obtained from Herbal Garden, Patanjali Research Institute, Haridwar, India. Individual voucher numbers were assigned to plants identified at the Council of Scientific and Industrial Research—National Institute of Science Communication and Information Resources (CSIR—NISCAIR) (Government of India), Delhi, India (Table 1). EC powder MMA (Colorcoat EC4S-B) was sourced from Corel Pharma Chem (Gujrat, India). HPLC standards for Gallic acid (Potency-97.3%), Rutin (Potency-98.70%), and Ellagic acid (Potency-99.60%) were purchased from Sigma-Aldrich (St. Louis, MO, USA). Imperatorin (Potency-97.06%) was purchased from Otto Chemie Pvt. Ltd. (Maharashtra, India). Sodium chloride, ammonium nitrate, potassium phosphate, potassium chloride, potassium citrate (monohydrate), uric acid, lactic acid sodium salt (Sodium DL-lactate), Porcine gastric mucin-type II, hydrochloric acid, pepsin, calcium chloride (dihydrate), porcine bile salts, Phorbol 12-myristate 13-acetate (PMA), and *Escherichia coli* derived lipopolysaccharide (LPS) were purchased from Sigma-Aldrich (St. Louis, MO, USA). Porcine Pancreatic Lipase (PL) were purchased from MP Biomedicals (Rue Geiler de Kaysersberg, Illkirch-Graffenstaden, France). Dulbecco’s Modified Eagle Medium (DMEM), Fetal Bovine Serum (FBS), Antibiotics, Trypsin-EDTA, Verso cDNA synthesis kit and The PowerUp SYBR Green Master Mix were purchased from Thermo Fisher Scientific (Waltham, MO, USA). RNeasy mini kit was procured from Qiagen (Hilden, Germany). Calcein AM dye was purchased from Cayman chemical (Ann Arbor, MI, USA). Human recombinant TNF-α was purchased from PeproTech (Thermo Fisher Scientific (Waltham, MO, USA). Soluble IL-6 and TNF-α specific human OptEIA™ ELISA kits were purchased from BD Biosciences (Franklin Lakes, NJ, USA).

### 5.2. Preparation of Cologrit Formulation and Tablet

Cologrit powder (Batch no. PRFT/CHIN/0422/0196) was created by combining different quantities of the plant extracts listed in Table 1. The fruit of *Aegle marmelos* (L.) Corrêa and the bark of *Holarrhena antidysenterica* (L.) Wall. ex A. DC were ground separately, and then sieved through a #16 mesh sieve. *Cuminum cyminum* L. seeds, *Trachyspermum ammi* (L.) Sprague fruit, and *Foeniculum vulgare* Mill. fruit were mild-roasted individually for 30 min at 60 °C, then ground and sieved through a #16 mesh size sieve. In the formulation, a hydromethanolic [40% H_2_O and 60% CH_3_OH] extract of *Cinnamomum camphora* (L.) J. Presl leaf was also applied. All components were combined and blended for 15 min at 25 rpm in an octagonal blender into Cologrit formulation.

Cologrit tablets (Batch No. CHIH/COLA/0222/2320) were prepared by blending Cologrit powder with microcrystalline cellulose in an octagonal blender for 15 min at 25 rpm (Model: OB 25 L, Kevin process technologies Pvt. Ltd., Gujrat, India). The mixture was then mixed with excipients at the concentrations listed in Table 2. Granules were made in a fluidized bed dryer (Model: SC-12, Kevin process technologies Pvt. Ltd., Gujrat, India) and tablets were prepared on a 16-station rotary tableting machine with pre-compression (Model: FM-04, Falcon machineries, Gujarat, India). The tablet compression and storage were carried out in a conditioned room temperature of 25 ± 2 °C.

### 5.3. Enteric Coating of Cologrit Tablets

For 6% (*w*/*w*) EC, appropriate weight of MMA was dissolved in isopropyl alcohol (IPA) for 10 min while stirring at 200 rpm. Dichloromethane (DCM) was added to the prepared solution (Ratio of IPA: DCM set at 60:40), and stirred for another 30 min at 300 rpm [78]. The prepared solution was filtered and used to coat the Cologrit tablets.

### 5.4. Physical Evaluation of Tablet

A digital weighing balance (Model: CG 302, Aczet Pvt. Ltd., Mumbai, Maharashtra, India) was used to determine the average weight of 20 tablets. Average friability of 20 tablets at 25 rpm for 4 min was measured using test apparatus (Model: 903, Electronics India, Panchkula, Haryana, India). A hardness tester (Labotech, Ambala cantt, India) was used to determine the average hardness of ten tablets. A Vernier Calliper (Model: 532–119, Mitutoyo, Kawasaki, Japan) was used to measure the thickness and diameter of the tablets.

### 5.5. Simulated Gastrointestinal (GIT) Digestion Assay

EC Cologrit tablets (11 × 500 mg) were suspended into 20 mL of MilliQ water for the study. As a negative control, only water without the Cologrit tablet was used. Simulated digestion of the Cologrit tablets was done following the protocol mentioned by Deloid et al. [47]. In the mouth phase, 20 mL of the water w/wo Cologrit EC tablets were mixed with 20 mL of simulated saliva fluid (pH 6.8, bolus) containing NaCl (50 mM), NH_4_NO_3_ (80 mM), KH_2_PO_4_ (9 mM), KCl (5 mM), K_3_C_6_H_5_O_7_ 2 mM, C_5_H_4_N_4_NaO_3_ 0.2 mM, Urea 0.6 mM, Sodium DL Lactate (2.6 mM), mucin (3.0 mg/mL) and amylase (150 unit/mL), and vigorously shaken for 15 s.

In the stomach phase, 20 mL of mouth phase bolus were mixed with 20 mL of simulated gastric fluid (chyme) containing NaCl (684 mM), HCl (72 mM) and pepsin (3.2 mg/mL), adjusted to pH 2.5, and incubated at 37 °C for 2 h on a shaker (100 rpm). 30 mL of stomach phase chyme with pH adjusted to 7.0 was mixed with small intestinal fluid (NaCl (150 mM) and CaCl_2_ (10 mM), bile salt (5.0 mg), and lipase (60 mg/mL; digestae) for 2 h at 37 °C with constant stirring. The pH of the solution was kept at 7.0 by adding 10 mL of 0.1 N NaOH solution. The dissolved Cologrit concentration in the final digestae was 11.06 mg/mL. All the digestae were stored at −80 °C and used for biochemical analysis.

### 5.6. HPLC-Based Phytochemical Analysis

Individually, Gallic acid, Rutin, Ellagic acid, and Imperatorin standards were dissolved in methanol to make 1000 ppm standard stock solutions. Imperatorin was further diluted for preparing a 100 ppm working standard solution. All the other standards were diluted to a working standard solution of 25 ppm. 250 mg of Cologrit was sonicated for 30 min in 10 mL of water: methanol (20:80) mixture. The mixture was centrifuged at 10,000 rpm and filtered through a 0.45 µm nylon filter. HPLC analysis was performed on the standards’ filtrate, and Cologrit tablet containing chyme and digestae samples collected under sub-Section 5.5.

HPLC was performed using Prominence-XR HPLC system (Shimadzu, Japan) equipped with Quaternary pump (Nexera XR LC-20AD XR), DAD detector (SPD-M20 A), Auto-sampler (Nexera XR SIL-20 AC XR), Degassing unit (DGU-20A 5R) and Column oven (CTO-10 AS VP). Separation was achieved using a Shodex C18-4E (5 µm, 4.6×250 mm) column subjected to binary gradient elution. The two solvents used for the analysis were water containing 0.1% acetic acid in water (solvent A) and acetonitrile (solvent B) set in a gradient program (Time (T) = 0: A% = 100%, B = 0%; T = 5: A%= 95%, B = 5%; T = 10: A% = 90%, B = 10%; T = 50: A% = 70%, B = 30%; T =60: A% = 45%, B = 55%; T =70: A% = 20%, B = 80%; T =71: A% = 100%, B = 0%; T =75: A% = 100%, B = 0%). Column temperature was kept 30 °C and flow was set 1.0 mL/min during the analysis. 10 µL of standard and test solutions were injected. Wavelength was set 253 nm (for Rutin, Ellagic acid, and Imperatorin) and 270 nm (for Gallic acid).

### 5.7. UV Spectroscopy Analysis

During the digestion phase, chyme and digestae samples were collected every 20 min, and analyzed for Cologrit tablet dissolution using UV-1800 spectrometer (Shimadzu, Kyoto, Japan). A full spectrum UV scan (200–1000 nm) was performed using the samples and Cologrit specific absorbance spectra λmax were determined at 339 nm, with no optical interferences in the presence of biomolecules. Spectragryph software (Software-Entwicklung, Oberstdorf, Germany) was used to spectra and data processing.

### 5.8. Cell Culture

Caco-2, HT-29, THP-1 and U937 cells were obtained from National Centre for Cell Science, Maharashtra, India (local repository for American Type Culture Collection). THP-1-Blue NFκB reporter cell was purchased from InvivoGen (San Diego, CA, USA). HT-29, and Caco-2 cells were grown in high-glucose DMEM media supplemented with 10% heat-inactivated Fetal Bovine Serum (FBS), 10 mM HEPES buffer, 100 IU/mL Penicillin, 100 µg/mL Streptomycin and 1% non-essential amino acids. U937, THP-1 and THP-1-Blue NFκB reporter cells were grown in normal RPMI-1640 media supplemented with 10% FBS, 10 mM HEPES buffer, 100 IU/mL Penicillin, 100 µg/mL Streptomycin and 1% non-essential amino acids.

### 5.9. Digestae Preparation for Cell-Based Assays

For the cell-based studies, fresh digestae containing 11.06 mg/mL of enzyme digested Cologrit tablet were prepared using the protocol described in sub-Section 5.5. Water only digestae was considered as untreated control. All of the digestae were adjusted to pH 7.0 and diluted in DMEM media (without supplements) in a 1:3 ratios for making stock solution. As required by the assay, additional dilutions were prepared in incomplete DMEM media.

### 5.10. Dose Response Analysis

Cologrit was suspended in 2% FBS containing DMEM at the concentrations of 1, 3, 10, 30, and 100 μg/mL. Following pre-incubation, HT-29 cells were plated at the concentration of 2 × 10^4^ cells/well in 96 well plates and incubated overnight. Next day, the cells were treated with different concentrations of Cologrit for 24 h. Caco-2 cells were plated at the concentration of 1 × 10^4^ cells/well in 96 well plates and pre-incubated overnight. Digestae w/wo Cologrit tablets were dissolved in serum-free DMEM medium at concentrations of 1, 3, 10, 30, and 100 μg/mL. In parallel, Caco-2 cells following pre-incubation were also treated for 24 h with 0.65 percent, 1.15 percent, 2.5 percent, and 5 percent (*w*/*v*) Dextran Sodium Sulfate (DSS) colitis grade (Mol. Wt. 36000–50000; MP Biomedicals, LLC, Illkirch, France) solution prepared in serum-free medium. At end of treatment time, media was removed and the cells were washed with sterile PBS. Cells were incubated for 3 h with a 10 μg/mL concentration of Alamar blue™, and fluorescence was measured at Ex. 560 nm/Em. 580 nm using Envision multimode plate reader (PerkinElmer, Waltham, MA, USA).

### 5.11. ELISA-Based Pro-Inflammatory Cytokines Analysis

For the IL-6 and TNF-α release estimation using ELISA assay, 1 × 10^5^ THP-1 cells/mL were differentiated using 20 ng/mL PMA overnight. The next day, the PMA-containing media was replaced with fresh complete cell culture media, and the differentiated cells rested for 4 days with intermittent media change. Pre-treatment of Cologrit was given to the differentiated cell at the concentrations of 3, 10, 30 and 100 µg/mL for 24 h. Following day, the cells were co-treated with LPS (100 ng/mL) and varying concentrations of Cologrit for 6 h. Cell soup was collected and used for the quantification of cytokines according to manufacturer instructions.

### 5.12. Evaluation of NFκB Response

THP-1-Blue NFκB reporter cells were plated in 96-well plates at a seeding density of 5 × 10^5^/mL and pre-incubated overnight. Cells were subsequently co-treated for 24 h with 10 ng/mL TNF-α and Cologrit at 10, 30, and 100 µg/mL concentrations. The expression of NFκB was measured as Secreted Embryonic Alkaline Phosphatase (SEAP) and quantified using the QUANTI-Blue assay (InvivoGen, CA, USA) according to the manufacturer’s instructions. The optical density at 630 nm readout was determined using an Envision multimode plate reader (PerkinElmer, Waltham, MA, USA).

### 5.13. RNA Isolation and RT-qPCR Analysis

HT-29 cells were seeded at a density of 2 × 10^5^/well in 12 well plates and incubated overnight. Cells were pre-treated with/without Cologrit at the varying concentrations of 10, 30 and 100 µg/mL for 24 h. Cells were subsequently co-treated for 24 h with 10 ng/mL TNF-α and Cologrit at 10, 30, and 100 µg/mL concentrations. Qiagen’s RNeasy mini kit was used to extract total RNA from the treated cells. 1 µg of RNA was reverse-transcribed into cDNA using the Verso cDNA synthesis kit according to the manufacturer’s instructions. The mRNA expression was analyzed using human primer sequences of IL-8: Fw- 5′ GCTTGAAGTTTCACTGGCATCT 3′; Rv- 5′ AGTTTTTGAAGAGGGCTGAGAAT 3′; CXCL5: Fw- 5′ AGCTGCGTTGCGTTTGTTTAC 3′; Rv- 5′ TGGCGAACACTTGCAGATTAC 3′; NFκB: Fw- 5′ GTGGTGCCTCACTGCTAACT 3′; Rv- 5′ GGATGCACTTCAGCTTCTGT 3′, and IL-1β: Fw- 5′ GTCGGAGATTCGTAGCTGGA 3′; Rv- 5′ ATGATGGCTTATTACAGTGGCAA 3′; and β-actin (housekeeping gene): Fw- 5′ CATGTACGTTGCTATCCAGGC 3′; Rv-5′ CTCCTTAATGTCACGCACGAT 3′. For RT-PCR analysis, a mixture containing 5 µL of PowerUp SYBR Green Master Mix, 5 ng of cDNA template, and 0.5 µL (200 nM) forward and reverse primers was prepared. qRT-PCR was performed using qTOWER3G (Analytik-Jena, Jena, Germany) instrument set at conditions- 95 °C for 5 min, 95 °C for 30 s, and 55 °C for 30 s for 40 cycles, with a melt curve obtained for each sample. The obtained mRNA expression values were normalized to β-actin, and relative mRNA expression was calculated using the 2^−Ct^ technique. Obtained expression values were normalized against β-actin and relative mRNA expression was quantified using the 2^−∆∆Ct^ method [79].

### 5.14. Monocyte Adhesion Assay

HT-29 cells were plated at a density of 5 × 10^4^ cells/well in 96-well plates. Pre-treatment of the HT-29 cells was given with Cologrit at the concentrations of 3, 10, 30, and 100 µg/mL for 24 h. Next day, cells were co-treated with TNF-α (10 ng/mL) and varying concentrations of Cologrit for 24 h. U937 cells were pre-labelled with Calcein AM dye (0.5 µM) for 30 min at 37 °C. Subsequently, HT-29 cells were co-incubated with the pre-labelled U937 cells (1 × 10^5^ cells/well) for 2 h. Non-adherent cells were washed with 1 × PBS, and fluorescence was measured with using Infinite 200Pro multimode plate reader (Tecan, Zurich, Switzerland) at Ex. 485 nm and Em. 530 nm.

### 5.15. Prophylactic Treatment with Cologrit Digestae

Caco-2 cells were pre-incubated with Cologrit digestae doses of 1, 3, 10, 30, and 100 µg/mL prepared in incomplete DMEM medium for 24 h. The cells were subsequently co-treated with 2.5% percent DSS and varying concentrations Cologrit digestae prepared in serum-free cell culture media. As negative and positive controls, water-based digestae w/wo 2.5 percent DSS were used. The treated cells were incubated for 24 h and then washed with PBS. The cells were treated with a 10 µg/mL of Alamar blue^TM^ for 3 h. Fluorescence was measured at Ex. 560 nm and Em. 580 nm using Envision multimode plate reader (PerkinElmer, Waltham, MA, USA).

### 5.16. Oxidative Stress Analysis

Caco-2 cells were pre-treated with the Cologrit digestae concentrations of 1, 3, 10, 30, and 100 μg/mL prepared in serum-free DMEM media for 24 h. Next day, cells were co-treated with 2.5% DSS and varying concentrations of Cologrit digestae prepared in serum-free media for 24 h. Water based digestae w/wo 2.5% DSS were treated as negative and positive controls, respectively. At the end of incubation time, cells were washed with 10 µg/mL (100 μL) of 2′, 7′-dichlorofluorescein diacetate dye and incubated in dark for 45 min at 37 °C. Fluorescence was measured at Ex. 490 nm and Em. 520 nm using the Envision multimode plate reader (PerkinElmer, Waltham, MA, USA).

### 5.17. Statistical Analysis

GraphPad Prism 7 was used for statistical analysis (GraphPad Software, San Diego, CA, USA). The data were presented as mean ± standard deviation (SD). Significance between distinct treatment groups was determined using one-way ANOVA, followed by Dunnett’s posthoc analysis. At a *p*-value of <0.05, the results were deemed statistically significant.

## Figures and Tables

**Figure 1 pharmaceuticals-16-00063-f001:**
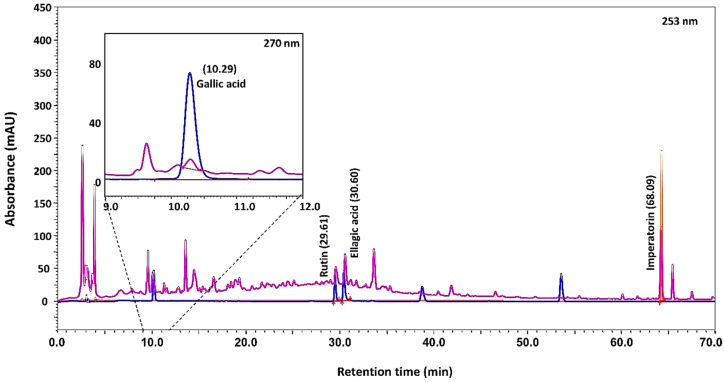
Phytochemical composition of the Cologrit tablet powder. HPLC based analysis showed the presence of Gallic acid (retention time 10.29 min), Rutin (retention time 29.61 min), Ellagic acid (retention time 30.60 min), and Imperatorin (retention time 68.09 min). Table 2 displays phytochemical concentration of tablet.

**Figure 2 pharmaceuticals-16-00063-f002:**
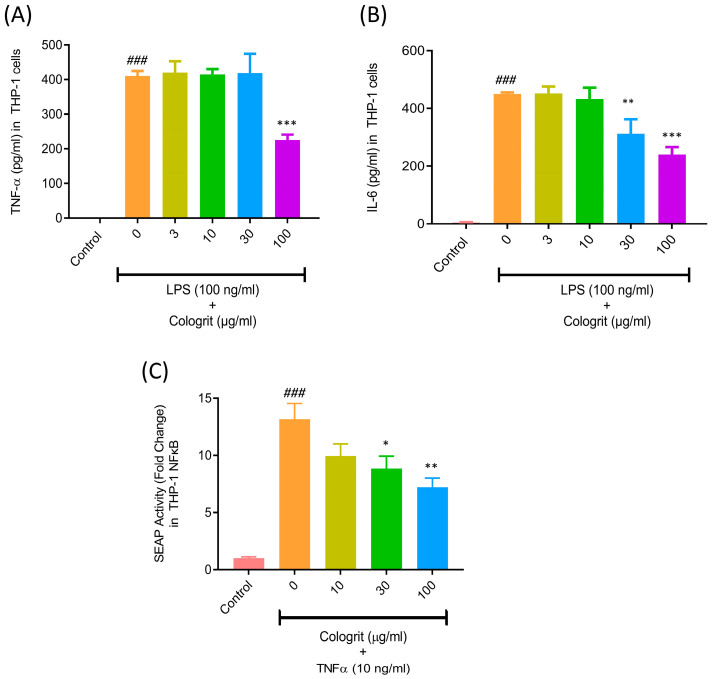
Anti-inflammatory potential of Cologrit formulation in THP-1 cells. Anti-inflammatory activity of the Cologrit formulation was investigated in PMA-transformed THP-1 cells stimulated with LPS (100 ng/mL) through analysis of solubilized pro-inflammatory cytokines (**A**) TNF-α and, (**B**) IL-6. (**C**) NFκB activation and its amelioration by Cologrit formulation was analysed in TNF-α-stimulated THP-1-Blue NFκB reporter cells. All the experiments were performed thrice in triplicate. Statistical analysis was performed using one-way ANOVA followed by Dunnett’s posthoc test. ### *p*-value < 0.001 (Control *versus* (vs.) LPS (100 ng/mL) treatment; * *p*-value < 0.05 (LPS treatment vs. Cologrit + LPS treatment; ** *p*-value < 0.01 (LPS treatment vs. Cologrit + LPS treatment; *** *p*-value < 0.001 (LPS treatment vs. Cologrit + LPS treatment).

**Figure 3 pharmaceuticals-16-00063-f003:**
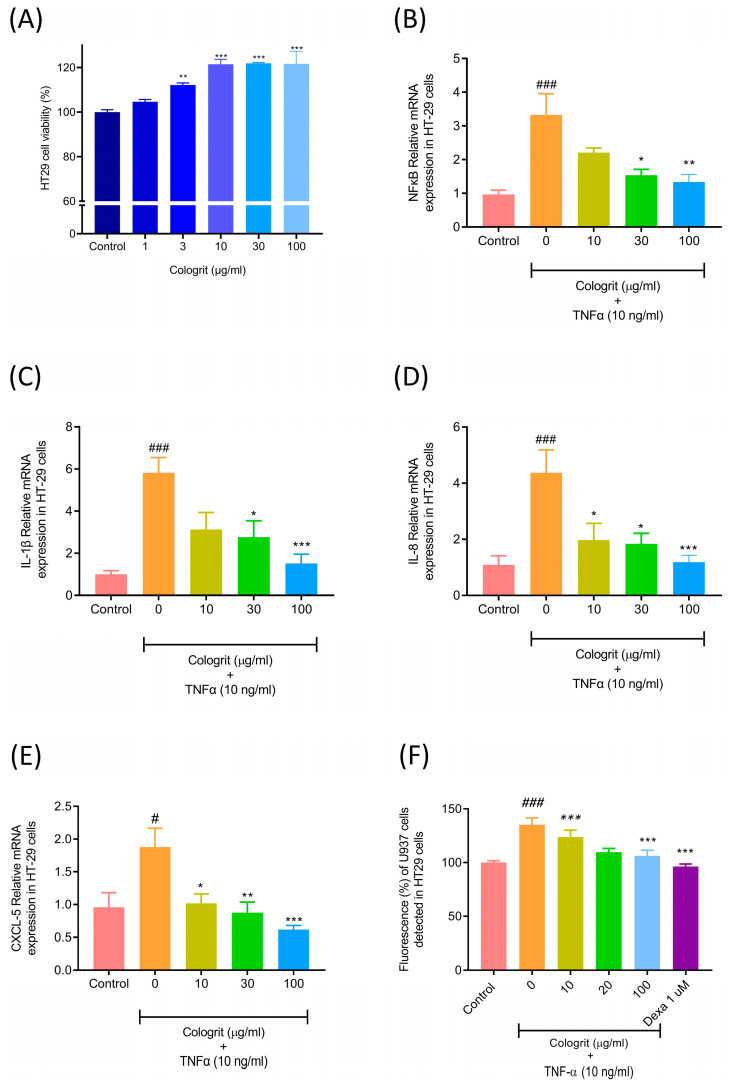
Biocompatibility and anti-inflammatory potential of Cologrit formulation in HT-29 cells. (**A**) Cell viability analysis was used to determine the dose response of the Cologrit formulation in HT-29 cells. The anti-inflammatory activity of Cologrit formulation was investigated in HT-29 cells stimulated with TNF-α (10 ng/mL) and pre- and co-treated with Cologrit formulation by measuring mRNA expression of (**B**) NFκB, (**C**) IL-1β, (**D**) IL-8, and (**E**) CXCL5. (**F**) The effect of TNF-α stimulation on adhesion molecule formation on the surface of HT-29 cells, as well as its amelioration by Cologrit, was studied by measuring fluorescence intensity of Calcein-AM dye-tagged human myeloid leukemia (U937) cells attached on the surface of HT29 cells. All the experiments were performed thrice in triplicate. Statistical analysis was performed using one-way ANOVA followed by Dunnett’s posthoc test. (**A**) ** *p*-value < 0.01; *** *p*-value < 0.001 (Control versus (vs.) treatment). (**B**–**F**) # *p*-value < 0.05 (Control vs. TNF-α (10 ng/mL) treatment; ### *p*-value < 0.001 (Control vs. TNF-α (10 ng/mL) treatment; * *p*-value < 0.05 (TNF-α (10 ng/mL) treatment vs. Cologrit + TNF-α (10 ng/mL) treatment; ** *p*-value < 0.01 (TNF-α (10 ng/mL) treatment vs. Cologrit + TNF-α (10 ng/mL) treatment; *** *p*-value < 0.001 (TNF-α (10 ng/mL) treatment vs. Cologrit/ Dexa (1µM) + TNF-α (10 ng/mL) treatment).

**Figure 4 pharmaceuticals-16-00063-f004:**
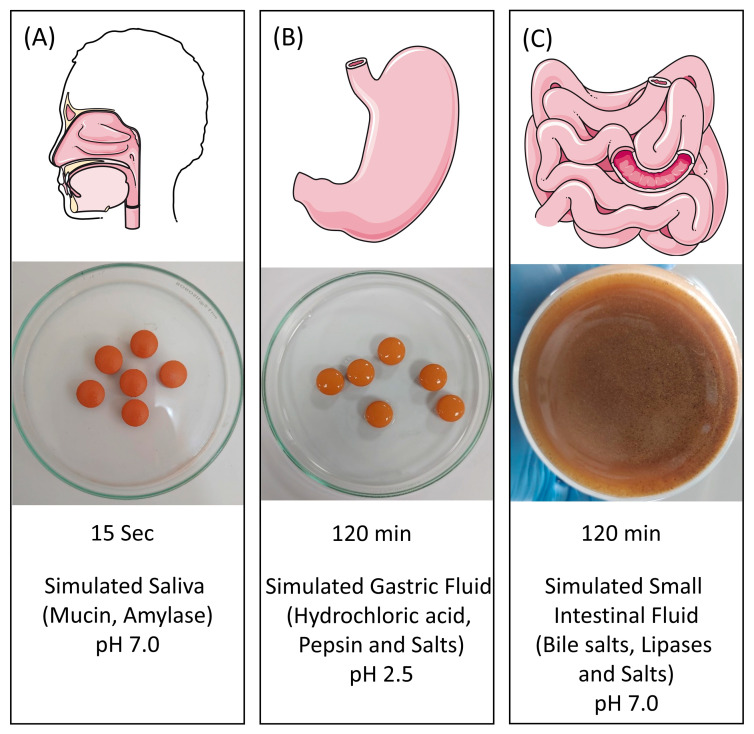
In vitro enzymatic digestion of EC Cologrit tablets in simulated gut model. Methacrylic acid-ethyl acrylate copolymer (1:1) type a (MMA) coated tablets of the Cologrit formulation were tested for solubility in the (**A**) mouth phase (pH 7.0) for 15 s, (**B**) stomach phase (pH 2.5) for 120 min, and (**C**) intestinal phase (pH 7.0) for 120 min, each with a different enzyme and ionic combination.

**Figure 5 pharmaceuticals-16-00063-f005:**
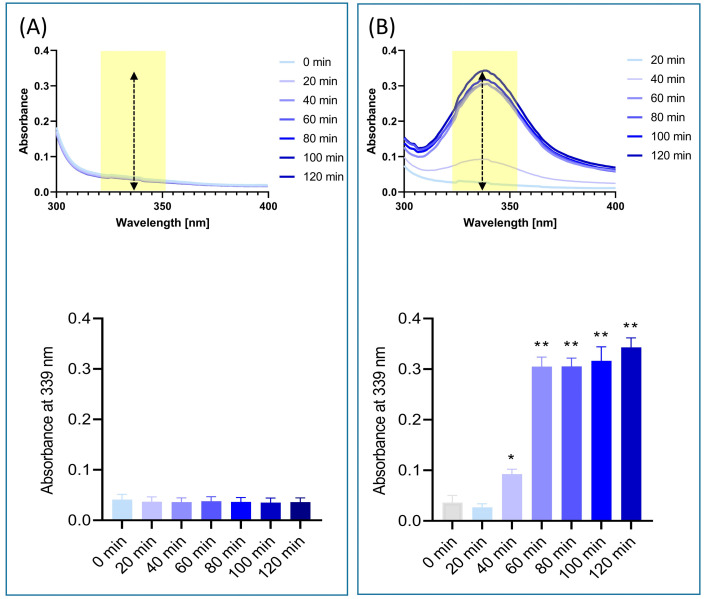
UV absorbance spectroscopy and phytochemical profiling study of in vitro digested EC Cologrit tablets. UV absorbance scan between the 300–400 nm range was done to validate the solubilization of the Cologrit tablet at various time-points. The solubilization of the tablets in (**A**) stomach phase, and (**B**) intestinal phase was evaluated using λmax analysis at 339 nm. (**C**) HPLC analysis of the 120 min Intestinal phase solubilized EC Cologrit tablets revealed the presence of Gallic acid (retention time 10.29 min), Rutin (retention time 29.61 min), Ellagic acid (retention time 30.60 min), and Imperatorin (retention time 68.09 min). Table 4 displays their phytochemical concentrations. All experimental conditions were repeated thrice. Statistical analysis was performed using one-way ANOVA followed by Dunnett’s posthoc test. * *p*-value < 0.05; ** *p*-value < 0.01 (0 min versus incubation treatment).

**Figure 6 pharmaceuticals-16-00063-f006:**
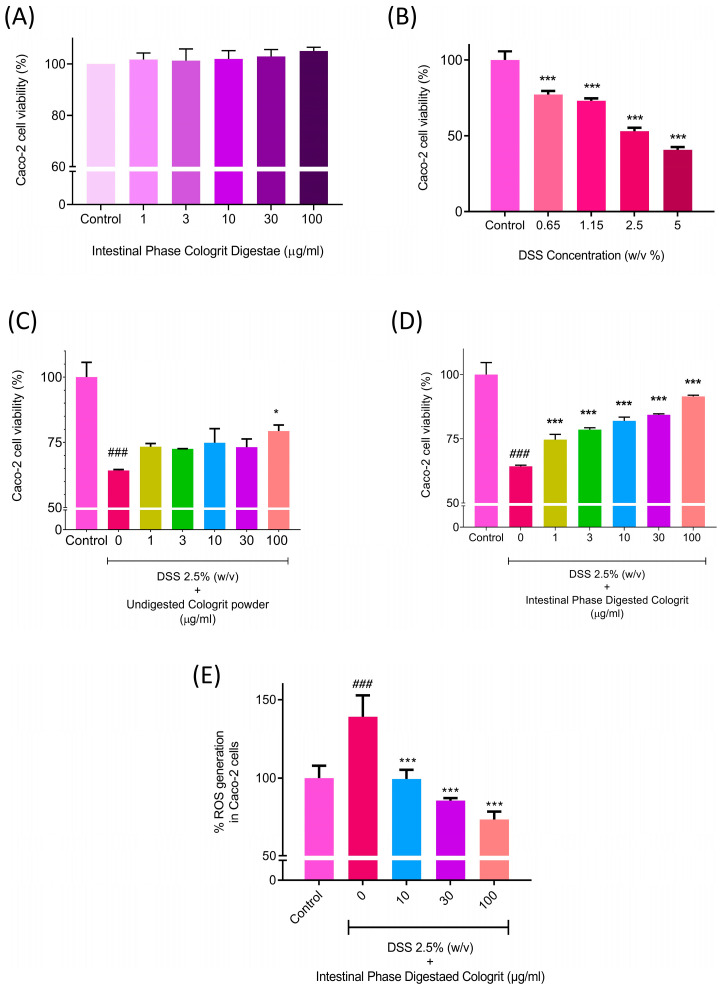
Biological activity of the intestinal phase solubilized EC Cologrit tablets. (**A**) Biocompatibility analysis of intestinal phase EC Cologrit tablet digestae done on Caco-2 cells, (**B**) dose range screening of dextran disodium sulfate (DSS) in Cac-2 cells performed using cell viability analysis, (**C**) protection against loss of cell viability by undigested EC Cologrit tablet in Caco-2 cells treated with 2.5% (*w*/*v*) DSS treatment, (**D**) protection against loss of cell viability by EC Cologrit tablet digestae in Caco-2 cells treated with 2.5% (*w*/*v*) DSS treatment, (**E**) protection against oxidative stress via ROS generation by EC Cologrit tablet digestae in Caco-2 cells treated with 2.5% (*w*/*v*) DSS treatment. Statistical analysis was performed using one-way ANOVA followed by Dunnett’s posthoc test. (**B**) *** *p*-value < 0.001 (Control versus (vs.) DSS treatment). (**C**–**E**) ### *p*-value < 0.001 (Control vs. DSS (2.5% *w*/*v*) treatment; * *p*-value < 0.05 (DSS (2.5% *w*/*v*) treatment vs. Cologrit + DSS (2.5% *w*/*v*) treatment; *** *p*-value < 0.001 (DSS (2.5% *w*/*v*) treatment vs. Cologrit + DSS (2.5% *w*/*v*) treatment).

**Table 1 pharmaceuticals-16-00063-t001:** Percentage of herbal extracts used in Cologrit formulation preparation.

Ingredients	Quantity (% *w*/*w*)	CSIR—NISCAIR (Voucher No.)
*Aegle marmelos* (L.) Corrêa	50%	NISCAIR/RHMD/Consult/2019/3453-54-27
*Holarrhena antidysenterica* (L.) Wall. ex A. DC	25%	NISCAIR/RHMD/Consult/2019/3453-54-110
*Cuminum cyminum* L.	10%	NISCAIR/RHMD/Consult/2019/3453-54-82
*Trachyspermum ammi* (L.) Sprague	5%	NISCAIR/RHMD/Consult/2019/3453-54-4
*Foeniculum vulgare* Mill.	5%	NISCAIR/RHMD/Consult/2019/3453-54-172
*Rosa indica* L.	3%	NISCAIR/RHMD/Consult/2022/3988-89-66
*Cinnamomum camphora* (L.) J. Presl	2%	NISCAIR/RHMD/Consult/2018/3134-83-75

**Table 2 pharmaceuticals-16-00063-t002:** HPLC analysis of EC Cologrit tablet dissolutions. Results represent Mean ± SD (N = 3). According to literature review, individual phytochemicals have been identified in *Aegle marmelos* (L.) Corrêa (Am); *Holarrhena antidysenterica* (L.) Wall. ex A. DC (Ha); *Cuminum cyminum* L. (Cc); *Trachyspermum ammi* (L.) Sprague (Ta); *Foeniculum vulgare* Mill. (Fv); *Rosa indica* L. (Ri); *Cinnamomum camphora* (L.) J. Presl (Cca), as cited in the specific references.

Sample Name	Gallic Acid (µg/mg)	Rutin (µg/mg)	Ellagic Acid (µg/mg)	Imperatorin (µg/mg)
Cologrit tablet powder	0.194	0.336	0.393	1.527
EC Cologrit tablet in Gastric Phase (t = 0 min)	ND	ND	ND	ND
EC Cologrit tablet in Gastric Phase (t = 60 min)	ND	ND	ND	ND
EC Cologrit tablet in Gastric Phase (t = 120 min)/Intestine Phase (t = 0 min)	ND	ND	ND	ND
EC Cologrit tablet in Intestine Phase (t = 60 min)	0.09 ± 0.06	0.25 ± 0.02	0.21 ± 0.04	0.04 ± 0.02
EC Cologrit tablet in Intestine Phase (t = 120 min)	0.12 ± 0.05	0.29 ± 0.007	0.28 ± 0.07	0.07 ± 0.02
Plants associated with detected phytochemicals [References]	Am [26,29,31] Ha [27] Cc [23,24,38] Ta [28] Fv [19,25] Ri [32] Cca [30,33]	Am [26,31] Ha [27] Cc [34] Ta [35] Cca [33]	Am [29,31] Cc [36] Fv [25]	Am [31,37] Fv [19]

**Table 3 pharmaceuticals-16-00063-t003:** Cologrit composition per 540 mg tablet.

Ingredients	Quantity (% *w*/*w*)
Cologrit powder	70.55
Gum acacia	13.88
Microcrystalline cellulose	0.64
Polyvinylpyrrolidone (PVP)	2.77
Corn starch	5.18
Isomalt	6.48
Talcum	0.09

**Table 4 pharmaceuticals-16-00063-t004:** Physical evaluation of EC Cologrit tablet.

Parameters	Results
Diameter (mm)	6.42–6.44
Thickness (mm)	10.82–10.91
Average wt. (mg)	542 ± 3.20
Wt. variation (mg)	537–549
Hardness (Kg/cm^2^)	6.96 ± 0.15
Friability (%)	<1

## Data Availability

The data presented in this study are available on request from the corresponding author. The data are not publicly available due to institutional policies.

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
