# Peer review of "Enteric-Coated Cologrit Tablet Exhibit Robust Anti-Inflammatory Response in Ulcerative Colitis-like In-Vitro Models by Attuning NFκB-Centric Signaling Axis"

_pharmaceuticals, 2022, doi:10.3390/ph16010063_

Round 1

Reviewer 1 Report

Congratulations on your accomplished and well-described research work. Ayurvedic medicine has many plant substances with interesting therapeutic effects to offer.

My comments relate to the following point:

1. on line 230-240 in fig.5 - the HPLC chromatogram indicates the presence (peak) of kaempferol (57.97), it is not included in the description, nor does it appear in table 4. (line 192 - 196) and in the description of the phytochemical analysis (489-491). Instead, there is coumarin, which is not indicated on the chromatogram, nor does it appear in Table 4. I hope this is an oversight, especially as flavonoids may also have an anti-inflammatory role.
2. in line 237 retention time with a capital letter, where the others are written in lower case.

Author Response

Dear Madam/ Sir,

We are deeply humbled by the encouraging comments and time put forward by the reviewer in improving the scientific quality of the manuscript. Based on the comments, we have now modified the manuscript and have made changes in track changes. Response to each comment/ suggestion has been added below in a point-wise manner.

We sincerely hope that these modifications will help in bringing the manuscript at par to the high-quality publications published in the peer-reviewed MDPI-Pharmaceuticals.

Thanking you.

Sincerely,

Anurag Varshney

Responses to Reviewer's Comments-

Congratulations on your accomplished and well-described research work. Ayurvedic medicine has many plant substances with interesting therapeutic effects to offer.

My comments relate to the following point:

  1. on line 230-240 in fig.5 - the HPLC chromatogram indicates the presence (peak) of kaempferol (57.97), it is not included in the description, nor does it appear in table 4. (line 192 - 196) and in the description of the phytochemical analysis (489-491). Instead, there is coumarin, which is not indicated on the chromatogram, nor does it appear in Table 4. I hope this is an oversight, especially as flavonoids may also have an anti-inflammatory role.

Response: We thank the reviewer for the keen observation. In the HPLC analysis, we had included a cocktail of several standards (blue line) including the Gallic acid, Rutin, Ellagic acid, Imperatorin, Kaempferol and Coumarin that have been standardized earlier based for the retention time using single entities. While the Gallic acid, Rutin, Ellagic acid, Imperatorin, Kaempferol and Coumarin were detected in the Cologrit through HPLC analysis (Table 4), the other two phytochemicals Kaempferol and Coumarin were not detected. Therefore, both of them have not been included in the table 4. Hence in the revised version of the manuscript, we have now omitted coumarin from the manuscript main text and Kaempferol from the HPLC chromatograms.

  1. in line 237 retention time with a capital letter, where the others are written in lower case.

Response: We thank the reviewer for the observation. We have now cross-checked throughout the manuscript and modified all to 'retention time'.

Reviewer 2 Report

This article provides an anti-inflammatory evaluation of a plant-based formula, besides its potential pharmaceutical formulation as an enteric coated tablet. I recommend accepting the article after considering the following comments.

1)    In tables 1-3, please omit the first column “Serial Number”.

2)    HPLC is a well-known term and no need to repeat the full un-abbreviated term many times.

3)    HPLC analysis for the determination of only four well-known and simple natural products is not enough. It would have been more favorable to carry out an LC-MS analysis of the Cologrit formula and search the spectrum against a library to identify several tens of natural products

4)    For the preparation of the enteric coating, why dichoromethane was added? please cite the source from which the method was adopted.

5)    In the experimental part, please provide the composition and concentrations of components of the simulated saliva, gastric, and small intestine fluids.

Author Response

Date: 24-12-2022

Dear Madam/ Sir,

We are deeply humbled by the encouraging comments and time put forward by the reviewer in improving the scientific quality of the manuscript. Based on the comments, we have now modified the manuscript and have made changes in track changes. Response to each comment/ suggestion has been added below in a point-wise manner.

We sincerely hope that these modifications will help in bringing the manuscript at par to the high-quality publications published in the peer-reviewed MDPI-Pharmaceuticals.

Thanking you.

Sincerely,

Anurag Varshney

Response to Reviewer Comments

This article provides an anti-inflammatory evaluation of a plant-based formula, besides its potential pharmaceutical formulation as an enteric coated tablet. I recommend accepting the article after considering the following comments.

  1. In tables 1-3, please omit the first column “Serial Number”.

Response: We thank the reviewer for the suggestion. We have now omitted all the serial numbers present in Tables 1-4.

  1. HPLC is a well-known term and no need to repeat the full un-abbreviated term many times.

Response: We thank the reviewer and based on the suggestion have used abbreviation of HPLC everywhere else other than the first time it was used in the manuscript main text.

  1. HPLC analysis for the determination of only four well-known and simple natural products is not enough. It would have been more favourable to carry out an LC-MS analysis of the Cologrit formula and search the spectrum against a library to identify several tens of natural products

Response: We completely agree with the reviewer's statement. Definitely, there would be tens of phytochemicals sourced from the different plant components of Cologrit that would be beneficial in an inflammatory disease like ulcerative colitis. Keeping this, we will definitely implement it and screen for all the detectable phytochemicals present in Cologrit using the LC-MS method and include them in our follow-up ulcerative colitis study to be performed in animal models. 

  1. For the preparation of the enteric coating, why dichoromethane was added? please cite the source from which the method was adopted.

Response: Use of Dichloromethane in the enteric coating was performed following the manufacturer's guideline. Other studies such as Sisodia et al., 2013, and Rahamathulla et al., 2021 have applied dichroloromethane and isopropyl alcohol in different ratios for the synthesis of Eudragit L 100 coated-tablets. Hence, in the manuscript we have updated the enteric coating process as-

Line 445: 4.3. Enteric coating of Cologrit tablets: For 6% (w/w) EC, appropriate weight of MMA was dissolved in isopropyl alcohol (IPA) for 10 minutes while stirring at 200 rpm. Dichloromethane (DCM) was added to the prepared solution (Ratio of IPA:DCM set at 60:40), and stirred for another 30 minutes at 300 rpm [78]. The prepared solution was filtered and used to coat the Cologrit tablets.

References:

Rathore, S. B. S., Sharma, A., Garg, A., Sisodiya, D. S. Formulation and evaluation of enteric coated tablet of Ilaprazole. (2013). International Current Pharmaceutical Journal. 2(10): 3329.

Begum, M. Y., Alqahtani, A., Ghazwani, M., Alhamood, N. A., Hani, U., Jajala, A., Rahamathulla, M. Development of Duloxetine Hydrochloride Tablets for Delayed and Complete Release Using Eudragit L 100. (2021) I. J. of Polymer Sci., 221:1-10.

  1. In the experimental part, please provide the composition and concentrations of components of the simulated saliva, gastric, and small intestine fluids.

Response: We thank the reviewer for the comment. However, we have included the composition and concentrations of the components of simulated saliva, gastric, and small intestine fluids in the 4. Materials and Methods section; 4.5 Simulated gastrointestinal (GIT) digestion assay:

Line 458-470: Simulated digestion of the Cologrit tablets was done following the protocol mentioned by Deloid et al. [47]. In the mouth phase, 20 ml of the water w/wo Cologrit EC tablets were mixed with 20 ml of simulated saliva fluid (pH 6.8, bolus) containing NaCl (50 mM), NH4NO3 (80 mM), KH2PO4 (9mM), KCl (5 mM), K3C6H5O7 2 mM, C5H4N4NaO3 0.2 mM, Urea 0.6 mM, Sodium DL Lactate (2.6 mM), mucin (3.0 mg/ ml) and amylase (150 unit/ ml), and vigorously shaken for 15 s.

In the stomach phase, 20 ml of mouth phase bolus were mixed with 20 ml of simu-lated gastric fluid (chyme) containing NaCl (684 mM), HCl (72 mM) and pepsin (3.2 mg/ ml), adjusted to pH 2.5, and incubated at 37 °C for 2 hours on a shaker (100 rpm). 30 ml of stomach phase chyme with pH adjusted to 7.0 was mixed with small intestinal fluid (NaCl (150 mM) and CaCl2 (10 mM), bile salt (5.0 mg), and lipase (60 mg/ ml; digestae) for 2 hours at 37 °C with constant stirring. The pH of the solution was kept at 7.0 by adding 10 ml of 0.1 N NaOH solution. The dissolved Cologrit concentration in the final digestae was 11.06 mg/ ml. All the digestae were stored at −80 °C and used for biochemical analysis.
